# A dark state of Chern bands:
# Designing flat bands with higher Chern number

Mateusz Łącki[1], Jakub Zakrzewski[1,2], Nathan Goldman[3*]

**1** Institute of Theoretical Physics, Jagiellonian University in Krakow, Łojasiewicza 11, 30-348 Kraków, Poland
**2** Mark Kac Complex Systems Research Center, Jagiellonian University in Krakow, Łojasiewicza 11, 30-348 Kraków, Poland
**3** CENOLI, Université Libre de Bruxelles, CP 231, Campus Plaine, B-1050 Brussels, Belgium
* Nathan Goldman ngoldman@ulb.ac.be

March 23, 2021

## Abstract

We introduce a scheme by which flat bands with higher Chern number $|C| > 1$ can be designed in ultracold gases through a coherent manipulation of Bloch bands. Inspired by quantum-optics methods, our approach consists in creating a "dark Bloch band" by coupling a set of source bands through resonant processes. Considering a $\Lambda$ system of three bands, the Chern number of the dark band is found to follow a simple sum rule in terms of the Chern numbers of the source bands: $C_D = C_1 + C_2 - C_3$. Altogether, our dark-state scheme realizes a nearly flat Bloch band with predictable and tunable Chern number $C_D$. We illustrate our method based on a $\Lambda$ system, formed of the bands of the Harper-Hofstadter model, which leads to a nearly flat Chern band with $C_D = 2$. We explore a realistic sequence to load atoms into the dark Chern band, as well as a probing scheme based on Hall drift measurements. Dark Chern bands offer a practical platform where exotic fractional quantum Hall states could be realized in ultracold gases.

# 1  Introduction

Designing Bloch bands with topological properties has become a central theme in the context of quantum-engineered systems [1–3]. In ultracold atoms, important efforts dedicated to the realization of emblematic topological lattice models [3,4], such as the Haldane [5,6] and the Harper-Hofstadter models [7–10], have recently led to the observation of a variety of topological phenomena, including quantized transport [9,11–14], chiral edge motion [14–18], vortex dynamics upon a quench [6,19], and quantized circular dichroism [20]. Of particular interest is the possibility of engineering flat bands with non-trivial Chern number, which could allow for the creation of strongly-correlated topological states reminiscent of fractional quantum Hall states [21]. While flat bands with Chern number $|C| = 1$ are reminiscent of the conventional Landau levels in the continuum [22], flat bands with higher Chern number $|C| > 1$ can lead to exotic strongly-correlated states that are specific to lattice systems [23–29]. However, such flat-band models remain unsuitable for cold-atom experiments, as they rely on the design of complicated multi-layered lattices or complex long-range hopping processes [21, 24, 25, 30–35].

In this work, we introduce a practical approach by which flat bands with large Chern number can be designed in ultracold gases, using realistic optical-lattice geometries. Inspired by the concept of "dark state" in quantum optics [36,37], our scheme relies on coupling a set of Bloch bands coherently in view of forming a non-degenerate "dark band". In the simplest $\Lambda$-scheme scenario, which involves three Bloch bands [Fig. 1], the Chern number of the engineered dark band is found to be dictated by a sum rule, $C_D = C_1 + C_2 - C_3$, where $C_{1,2,3}$ designate the Chern numbers associated with the three bare Bloch bands. Moreover, the flatness of the dark band is shown to be directly inherited from the bare bands. Altogether, this strategy offers a practical method for designing flat bands with predictable and controllable topological invariants in a wide range of cold-atom settings.

In the following, we discuss in detail the applicability and validity of this general approach in view of realizing flat bands with higher Chern number in available cold-atom setups. In particular, we describe a realistic method by which atoms can be loaded into the topological dark band. Besides, we present clear signatures of the prepared state based on Hall drift measurements [9,38–40]. While our results directly apply to cold-atom settings, our findings are general and could be relevant to light-induced topological phases in the solid state [41,42].

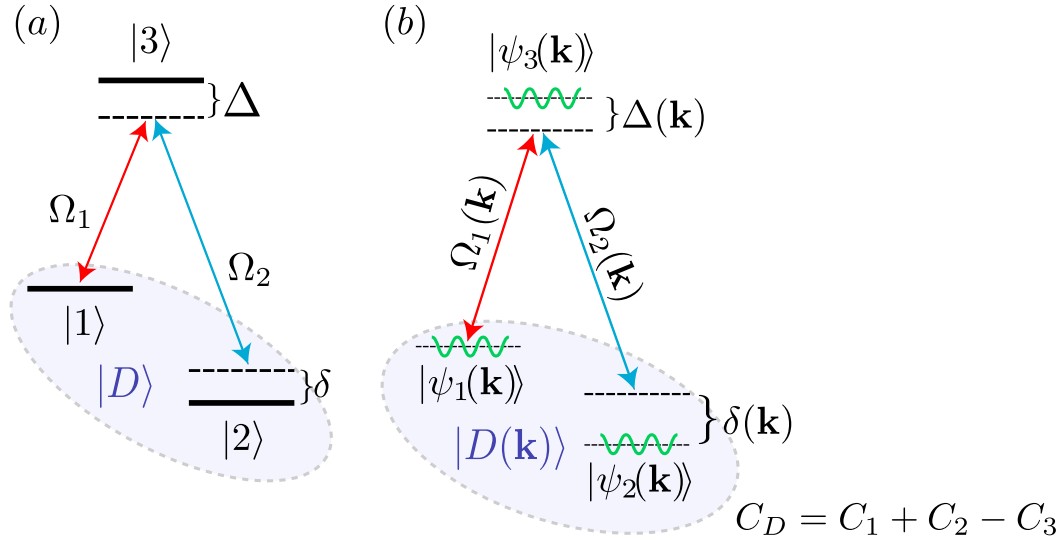

Figure 1: (a): The $\Lambda$ scheme involving three discrete levels. The Rabi couplings $\Omega_{1,2}$ and frequency detunings $\Delta$ and $\delta$ are indicated; see Eq. (2). A stable dark state $|D\rangle$ is created when $\delta \approx 0$; see Eq. (3). (b) The $\Lambda$ scheme involving three Bloch bands $E_{1,2,3}(\mathbf{k})$. The detunings and Rabi frequencies depend on the quasimomentum $\mathbf{k}$ of the Bloch states. The Chern number of the resulting "dark band" $C_D$ follows a simple sum rule in terms of the Chern numbers $C_{1,2,3}$ of the bare bands.

## 2 Dark state of a $\Lambda$ system

The dark-band scheme proposed and analyzed in this work is based on a simple three-band configuration, which is reminiscent of the well-known $\Lambda$ scheme of quantum optics [37]. As depicted in Fig. 1(a), a generic $\Lambda$ scheme involves three quantum states $|1\rangle, |2\rangle, |3\rangle$, with energies $E_{1,2,3}$, which are resonantly coupled by two driving fields. The Hamiltonian is written as

$$\hat{H}_{\Lambda,\text{lab}} = \hbar \begin{pmatrix} E_1 & 0 & 0 \\ 0 & E_2 & 0 \\ 0 & 0 & E_3 \end{pmatrix} + \hat{A}\sin(\omega_1 t) + \hat{B}\sin(\omega_2 t), \tag{1}$$

where $\hat{A}, \hat{B}$ are operators that describe the couplings. In a rotating frame, and upon the rotating wave approximation, the effective Hamiltonian takes the familiar form [37]

$$\hat{H}_{\Lambda} = \hbar \begin{pmatrix} 0 & 0 & \Omega_1^*/2 \\ 0 & \delta & \Omega_2^*/2 \\ \Omega_1/2 & \Omega_2/2 & -\Delta \end{pmatrix}, \tag{2}$$

where $\Omega_1^* = i\langle 1|\hat{A}|3\rangle$ and $\Omega_2^* = i\langle 2|\hat{B}|3\rangle$, and where $(\delta, \Delta)$ are the detunings illustrated in Fig. 1(a). For $\delta/\Omega_{1,2} \to 0$ and arbitrary $\Delta$, the dark state (DS),

$$|D\rangle = \frac{1}{2\bar{\Omega}} \left( \Omega_1|2\rangle - \Omega_2|1\rangle \right), \quad \bar{\Omega} = \frac{1}{2}\sqrt{|\Omega_1|^2 + |\Omega_2|^2}, \tag{3}$$

is an eigenstate of $\hat{H}_\Lambda$ corresponding to the eigenvalue $E_0 = 0$. The Hamiltonian $\hat{H}_\Lambda$ possesses two other "bright" eigenstates, with energies $E_\pm = \frac{\hbar}{2}(-\Delta \pm \sqrt{\Delta^2 + 4\bar{\Omega}^2})$ given by:

$$|B_\pm\rangle = N\left(\frac{\Omega_1^*}{E_\pm}|1\rangle + \frac{\Omega_2^*}{E_\pm}|2\rangle + |3\rangle\right). \tag{4}$$

In contrast with the dark state in Eq. (3), which forms a robust superposition of the two low-lying states ($|1\rangle$ and $|2\rangle$) only, the bright states involve all components [37].

We now apply this dark-state notion, which has been widely exploited in quantum optics [43–53] and atomic physics [54–60], to the case of three coherently coupled Bloch bands $E_{1,2,3}(\mathbf{k})$, where $\mathbf{k}$ denotes quasi-momentum.

## 3  $\Lambda$ system of Bloch bands and the sum rule

For the sake of concreteness, we consider the Bloch bands of the emblematic Harper-Hofstadter (HH) model [61], which can be engineered in optical-lattice experiments [7–10]. The corresponding Hamiltonian reads

$$\hat{H}_{\text{HH}} = -J \sum_{\mathbf{n}=(n,m)} \hat{a}_\mathbf{n}^\dagger \hat{a}_{\mathbf{n+1_y}} + \hat{a}_\mathbf{n}^\dagger \hat{a}_{\mathbf{n+1_x}} e^{i2\pi m\phi} + \text{h.c.}, \tag{5}$$

where $\mathbf{n} = (n, m)$ enumerates lattice sites on a 2D square lattice with lattice constant $a$, $J$ is the hopping amplitude, and where $a_\mathbf{n}/a_\mathbf{n}^\dagger$ annihilate/create a particle at site $\mathbf{n}$. The complex phase factor, which accompanies hopping along the $x$ direction and generates a uniform magnetic flux $2\pi\phi$ per plaquette, can be tuned in experiments [7–10].

Considering a flux of the form $\phi = 1/q$, with $q$ an odd integer, and applying periodic boundary conditions (PBC), the spectrum displays $q$ non-degenerate Bloch bands $\varepsilon_\nu(\mathbf{k})$, where $\nu = 1, \ldots, q$. We define the first Brillouin zone (BZ) as $\mathbf{k} \in [0, 2\pi/aq) \times [0, 2\pi/a)$. Each band is associated with a topological Chern number [62]

$$C_\nu = \frac{1}{2\pi} \int_{BZ} \mathcal{F} \, d^2k, \ \ \mathcal{F} = 2 \operatorname{Im}\langle\partial_{k_y}\psi_\nu(\mathbf{k})|\partial_{k_x}\psi_\nu(\mathbf{k})\rangle, \tag{6}$$

where $\psi_\nu(\mathbf{k})$ denotes an eigenstate in the $\nu$th band. For $q$ a generic odd integer, the Chern number of the central band $[\nu = (q+1)/2]$ reads $C_\nu = (-q+1)$, while $C_\nu = 1$ for all the other bands. Hence, except from the central band, the bands of the HH model are reminiscent of Landau levels (LL). In particular, these LL-like bands become flat in the limit $q \gg 1$.

We now discuss how a coherent coupling of these LL-like bands allow for the generation of flat bands with higher Chern number $|C| > 1$ within the HH model. This is achieved by building a $\Lambda$ system [Eq. (2)] from three selected bands $E_{1,2,3}(\mathbf{k})$ of the HH spectrum $\{\varepsilon_\nu\}$; we will denote the mean energy of each band as $\bar{E}_{1,2,3}$. In order to couple these bands quasi-resonantly, we consider a two-frequency drive with frequencies set to $\hbar\omega_{1,2} = \bar{E}_{1,2} - \bar{E}_3$; see Fig. 1(b). Assuming that the coupling field has a spatial periodicity that is compatible with the (magnetic) unit cell, the system forms a collection of decoupled $\Lambda$ systems, $\hat{H}_\Lambda(\mathbf{k})$, one at each quasi-momentum $\mathbf{k}$, and involving the three relevant states

$$|\psi_1(\mathbf{k})\rangle \equiv |1\rangle, \quad |\psi_2(\mathbf{k})\rangle \equiv |2\rangle, \quad |\psi_3(\mathbf{k})\rangle \equiv |3\rangle, \tag{7}$$

where $|\psi_n(\mathbf{k})\rangle$ denotes the Bloch state in band $n$ and quasimomentum $\mathbf{k}$; see Fig. 1(b). Applying the rotating wave approximation (RWA) [37], as detailed in Appendix B, and neglecting the other bands, this setting is indeed well described by Eq. (2), with momentum-dependent Rabi frequencies

$$\Omega_1^*(\mathbf{k}) = i\langle\psi_1(\mathbf{k})|\hat{H}_\Lambda(\mathbf{k})|\psi_3(\mathbf{k})\rangle, \quad \Omega_2^*(\mathbf{k}) = i\langle\psi_2(\mathbf{k})|\hat{H}_\Lambda(\mathbf{k})|\psi_3(\mathbf{k})\rangle, \tag{8}$$

and detunings $\delta(\mathbf{k}), \Delta(\mathbf{k})$. In this setting, one can define a dark state $|D(\mathbf{k})\rangle$ and two bright states $|B_\pm(\mathbf{k})\rangle$ at every quasi-momentum $\mathbf{k}$; the corresponding bands will be referred to as the "dark band" and the two "bright bands".

As a central result of this work, we find that the Chern number of the dark band is given by a simple sum rule:

$$C_D = C_1 + C_2 - C_3, \tag{9}$$

where $C_{1,2,3}$ denote the Chern numbers of the selected bands $E_{1,2,3}$. This simple rule is independent of: (i) the flux $\phi = 1/q$, (ii) the operator associated with the coupling field, and (iii) the chosen bands in the HH spectrum. Altogether, this provides a unique way to produce bands with predictable higher Chern number. We now provide the proof of Eq. (9) and discuss its regime of applicability.

The formula (9) is valid whenever an effective $\Lambda$ (three-band) configuration is achieved and the dark-state band is well separated from the two bright-state bands. To prove it, let us first remark that it is sufficient to show that the Chern number of the well-isolated bright bands is equal to $C_3$. Indeed, the couplings $\Omega_{1,2}$ only mix the three bands $E_{1,2,3}(\mathbf{k})$, so that $C_D + C_{B+} + C_{B-} = C_1 + C_2 + C_3$; together with

$$C_{B\pm} = C_3, \tag{10}$$

this implies the sum rule in Eq. (9). The hypothesis in Eq. (10) is demonstrated via the existence of a smooth deformation $|h(\mathbf{k}, t)\rangle$ that connects the bright states $|B_\pm\rangle = |h(\mathbf{k}, 1)\rangle$ to the bare state $|\psi_3(\mathbf{k})\rangle = |h(\mathbf{k}, 0)\rangle$. This deformation is provided by the expression

$$|h(\mathbf{k}, t)\rangle = N(\mathbf{k}, t)\left(\frac{t\Omega_1^*(\mathbf{k})}{E_\pm}|\psi_1\rangle + \frac{t\Omega_2^*(\mathbf{k})}{E_\pm}|\psi_2\rangle + |\psi_3\rangle\right), \quad t \in [0, 1], \tag{11}$$

$$= N(\mathbf{k}, t)\left(\frac{it}{E_\pm}|\psi_1\rangle\langle\psi_1|\hat{A} + \frac{it}{E_\pm}|\psi_2\rangle\langle\psi_2|\hat{B} + \hat{1}\right)|\psi_3\rangle, \tag{12}$$

where $N(\mathbf{k}, t)$ is a normalization factor, and where we used the definition of the Rabi frequencies $\Omega_{1,2}$ in Eq. (2). The deformation $|h(\mathbf{k}, t)\rangle$ in Eq. (11) is to be compared with the definition of the bright states in Eq. (4). Three observations are in order: (i) $|h(\mathbf{k}, t)\rangle \neq 0$ for all $t \in [0, 1]$, since at the very least the component $|\psi_3\rangle$ is non-zero; (ii) $|h(\mathbf{k}, t)\rangle$ is compatible with the BZ boundary conditions for all $t \in [0, 1]$; (iii) $|h(\mathbf{k}, t)\rangle$ acquires a global phase under a gauge transformation (this can be deduced from Eq. (12)). The Chern number of a band is preserved under such a homotopy, hence leading to the announced result in Eq. (10); this closes the proof of the sum rule in Eq. (9).

Importantly, the assumptions underlying the sum rule (9) impose a series of constraints on the chosen Bloch bands and system parameters, as we now explain.

First, the finite bandwidth of the bare bands $E_{1,2,3}(\mathbf{k})$ produces detunings $\delta(\mathbf{k})$ and $\Delta(\mathbf{k})$ in Eq. (2), which affects the flatness of the dark band and reduces the gap to the bright

bands; see Appendix C.1. This effect can be limited by noting that the width of the HH bands decreases exponentially with $q$ (except for the central band); see Appendix C.2.

Then, the Rabi frequencies $\Omega_{1,2}(\mathbf{k})$ should satisfy the inequalities $\delta(\mathbf{k}) \ll \Omega_{1,2}(\mathbf{k}) \ll |E_{\pm}(\mathbf{k})|$ to allow for a good separation of the dark band and optimize its flatness; this condition can already be reached for moderate $q \sim 7 - 10$. Besides, one should avoid pathological zeros $\bar{\Omega}(\mathbf{k}) = 0$, which would invalidate the dark state construction; see Appendix C.4.

Furthermore, for large $q \gg 1$, the HH spectrum forms a ladder of quasi-equally spaced Landau levels, which indicates that only special choices of bare bands $E_{1,2,3}$ can lead to a genuine $\Lambda$ configuration. In addition, the RWA is only valid when the resonant frequencies $\omega_{1,2}$ (and their differences $\Delta\omega$) are much larger than all other frequency scales, which also sets an important constraint on the chosen bands.

Finally, the unique bare band with $C_\nu \neq 1$ is the central band $[\nu = (q + 1)/2]$, for which the bandwidth to bandgap ratio is $O(1)$. This unfortunately rules out involving this band in our construction, which eventually implies the disappointing result $C_D = 1$ [Eq. (9)].

In order to overcome these limitations and constraints, we slightly generalize our scheme by introducing different atomic species (i.e. "spins"), as we describe in the next section.

Before doing so, we note that similar sum rules were numerically investigated in the multilayer configuration of Ref. [63]. The latter work considered the stacking of trivial (boron-nitride-type) and non-trivial (Haldane-type) 2D lattices, and it explored the total Chern number of the system at half-filling as a function of the inter-layer coupling strength. In that context, the calculated Chern number reflects the topology of a three-fold *degenerate* band, which is associated with the three underlying layers; using an open-system approach, this total Chern number was decomposed as a sum of sub-system indices [64]. We point out that the approach developed in the present work is radically different, as it involves the Chern number of a *non-degenerate* and well-isolated dark band; in particular, the sum rule in Eq. (9) directly relates the dark band's Chern number to the Chern numbers of the individual bare bands, without requiring the use of an open-system formalism.

# 4  The multi-species configuration

We now describe the multi-species configuration, which allows to produce dark bands with higher Chern number. Specifically, we propose to generate a flat and non-degenerate dark band with $C_D = 2$, by constructing a $\Lambda$ system made of two HH bands $[C_{1,2} = 1]$ and a trivial band $[C_3 = 0]$. The latter is provided by the lowest band of a square lattice without flux [Eq. (5) with $\phi = 0$, denoted $\hat{H}_0$], whose unit cell area $A_{\text{cell}} = qa \times a$ ensures a common BZ with $\hat{H}_{\text{HH}}$; the corresponding sites are located at $\tilde{\mathbf{n}} = (qn, m)$. This setting could be realized using different atomic (internal) states trapped in state-dependent potentials [66–68]; see Refs. [7, 69, 70] for schemes realizing state-dependent synthetic flux. In order to guarantee the validity of the RWA, we henceforth consider that each of the three selected bands $E_{1,2,3}$ is populated by a specific internal state $\sigma = \{1, 2, 3\}$. The $\Lambda$ coupling between the three bands is then performed by properly coupling the internal states with microwave fields. For convenience, we take the trivial band $(E_3)$ to be extremely flat (i.e. the corresponding state-dependent lattice is assumed to be very deep, with hopping amplitude $\tilde{J} \ll J$). We note that using only two internal states could also be envisaged in practice.

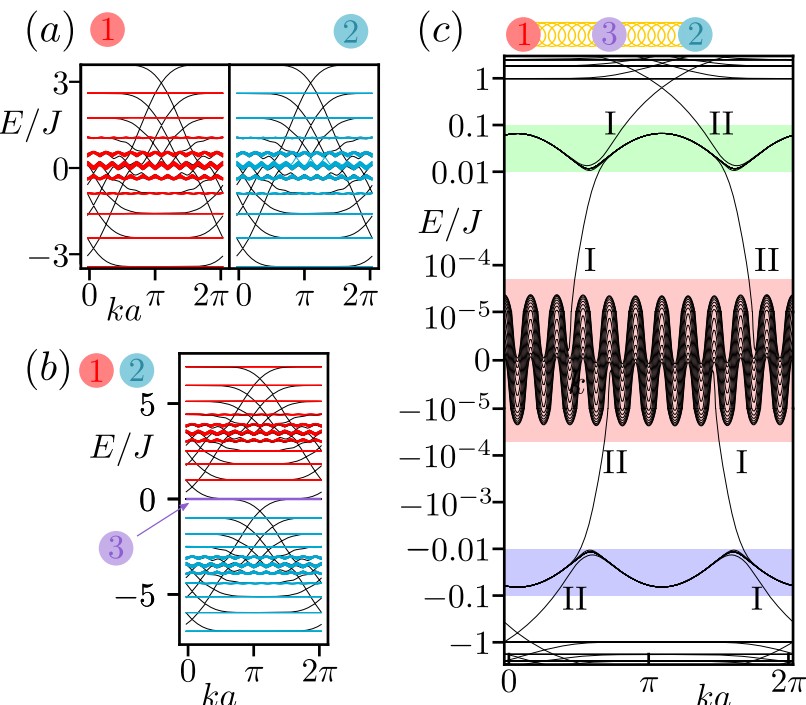

Figure 2: Designing a Λ system from two Hofstadter bands and a trivial flat band. (*a*) Hofstadter bands associated with two uncoupled atomic internal states ($\sigma = 1, 2$), for a flux $\phi = 1/11$. The black dispersions correspond to edge-state branches [65]; the system is diagonalized on a cylindrical geometry. (b) Adjusting the detunings $\delta_\sigma$ in $\hat{H}_\Lambda$ such that two Hofstadter bands (populated by states $\sigma = 1, 2$) become degenerate with a flat trivial band (populated by $\sigma = 3$) in the uncoupled limit $A_{1,2} = 0$. (c) Activating the coupling ($A_{1,2} \neq 0$, max $|\Omega_{1,2}(\mathbf{k})| = J/5$) splits the three overlapping bands into a flat dark band at zero energy (red shaded), and two bright bands (green/purple shaded). The flatness of the dark band is emphasized by a logarithmic scale. The edge-state branches, associated with the two edges (I,II) of the cylinder, indicate that the Chern number of the dark band is $C_D = 2$.

In a rotating frame, and upon applying the RWA, the Hamiltonian of this setting reads

$$\hat{H}_\Lambda = \sum_{\sigma=1}^{3} \left(\hat{H}_\sigma + \delta_\sigma \hat{P}_\sigma\right) + \frac{1}{2} \sum_{s=1,2} A_s \sum_{\tilde{\mathbf{n}}=(qn,m)} \hat{a}^\dagger_{\tilde{\mathbf{n}},s} \hat{a}_{\tilde{\mathbf{n}},3} + \text{h.c.}, \tag{13}$$

where $a^\dagger_{\tilde{\mathbf{n}},\sigma}$ creates an atom at site $\tilde{\mathbf{n}}$ in internal state $\sigma$; $\hat{H}_{1,2} = \hat{H}_{HH} \otimes \hat{P}_{1,2}$ and $\hat{H}_3 = \hat{H}_0 \otimes \hat{P}_3$, where $\hat{P}_\sigma$ projects onto the $\sigma$ component; $A_{1,2}$ denote the amplitudes of the coupling fields, and the detunings $\delta_\sigma$ are controlled by tuning the driving frequencies out of resonance. In particular, these detunings can be chosen such that two bands of the HH spectrum (populated by states $\sigma = 1, 2$) become degenerate with the trivial band (populated by $\sigma = 3$) in the decoupled limit $A_{1,2} = 0$; see Figs. 2(a)-(b). Upon activating the coupling ($A_{1,2} \neq 0$), these three bands split into the dark band and the two bright bands shown in Fig. 2(c). While the bright bands acquire a small dispersion, due to the $\mathbf{k}$-dependence of the Rabi frequencies $\Omega_{1,2}(\mathbf{k})$ in Eq. (8), the dark band at zero energy remains almost perfectly flat; see Appendix C.4.

One confirms that the Chern number of the dark band is $C_D = 2$, as dictated by the sum rule (9); this is readily obtained by analyzing the edge-state branches of opposite chirality that enter and leave the dark band in Fig. 2(c); see Ref. [65]. One also finds that the Chern number of the bright bands are zero [Eq. (10)], such that the total Chern number of the three coupled bands is indeed conserved. We have verified that these results are generic, in the sense that they do not depend on the specific form of the coupling operator.

## 5 Higher Chern number from centre-of-mass responses

The Chern number $C_D$ of the constructed dark band could be measured through different probes in ultracold atoms, such as center-of-mass responses [9], edge-state spectroscopy [71, 72], and circular dichroism [20]. Here, we validate our approach by simulating the center-of-mass displacement of an atomic cloud, loaded in the dark band, and perturbed by a linear potential gradient [9, 38–40].

As a first step, we numerically simulate the following protocol [38]: We initially confine the system, using sharp rectangular walls; we generate a dark band [as illustrated in Fig. 2(c)] in this geometry and completely fill it with non-interacting fermions; we remove the confining walls, and act on the particles with a weak linear potential gradient aligned along the $y$ direction, $\hat{F} = Fa \sum_{s=1,2,3} \sum_{\mathbf{n}=(n,m)} m \hat{n}_{\mathbf{n},s}$; finally, we calculate the time evolution of the particle density $\rho(x, y)$, and evaluate the center-of-mass displacement $\Delta x(t)$. The latter observable is related to the Chern number of the populated band ($C_D$) through the relation [9, 38–40]

$$\Delta x(t) = \frac{qa^2 F}{h} C_D t, \quad \text{for a flux } \phi = 1/q. \tag{14}$$

We show the simulated time-evolved density in Figs. 3(a)-(b), which demonstrate a clear transverse drift of the cloud (along the $x$ direction). We note that the density modulation along the $x$ direction reflects the coupling to the "trivial" lattice, of area $A_{\text{cell}} = qa \times a$, which supports the component $\sigma = 3$ (which is absent in the dark state). In Fig. 3(a),(b), the applied force is weak compared to the gap $\Delta_{\text{db}}$ separating the dark band from the bright bands, $Fa \ll \Delta_{\text{db}}$, such that the particles' motion adiabatically follows the dark band. In this linear-response regime, Eq. (14) is applicable [38, 39], and we extract an "experimental" value

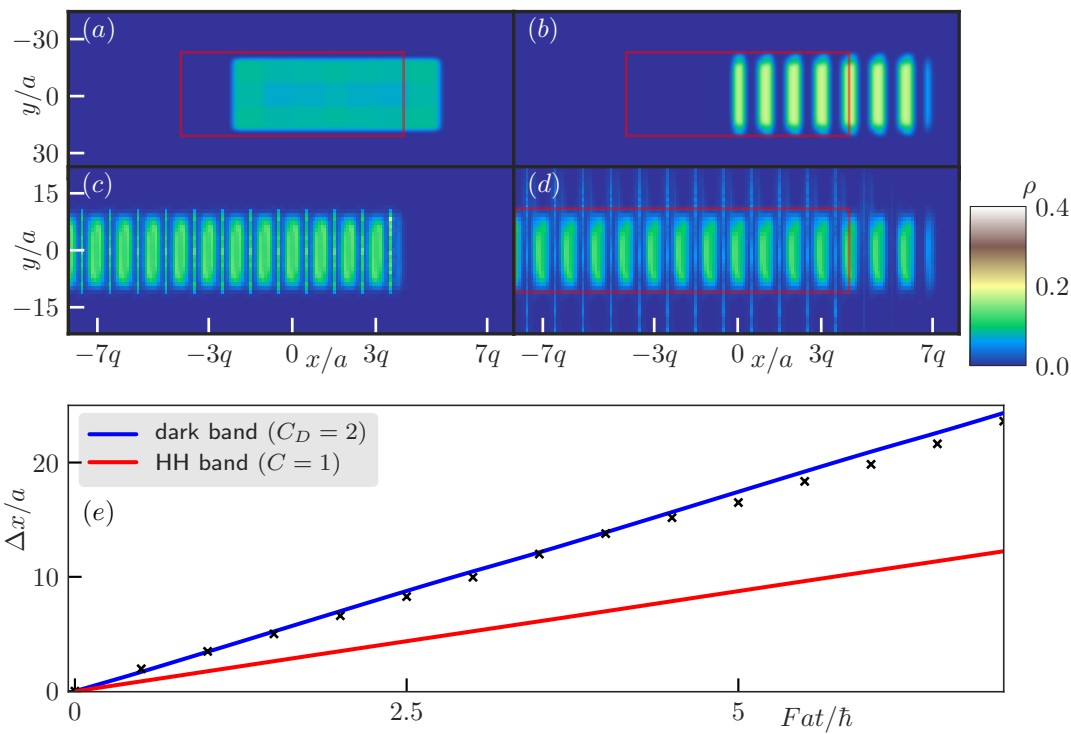

Figure 3: Hall drift in the dark band. (a) Time-evolved density $\rho(x,y)$ of non-interacting fermions populating a regular Hofstadter band at flux $\phi = 1/11$ (with Chern number $C=1$), initially confined within a small region (red rectangle) and then exposed to a linear gradient along $y$ upon release. The evolution duration is $t = 10(\hbar/Fa)$, the force strength $F = 5 \cdot 10^{-4} J/a$, and the system size $88 \times 44$. (b) Same as in (a) but instead of a regular Hofstadter band, one initially populates a dark state band with $C_D = 2$; see Fig. 2 and related text. The force strength $F$ is the same as in $(a)$, and is weak compared to the gap, $\Delta_{\mathrm{db}} = 8 \cdot 10^{-2} J$, hence allowing for an accurate measurement of the Chern number $C_D^{\mathrm{exp}} = 1.99$ through the center-of-mass drift [panel (e) and Eq. (14)]. (c) Density at the end of the proposed loading sequence, using a total ramp duration $T = 3 \cdot 10^3 \hbar/J$ and system size $242 \times 22$. (d) Time-evolved density of the realistically prepared state, when exposed to the linear gradient along $y$ for the same duration as in (a)-(b), $t = 10(\hbar/Fa)$; here we set $F = 5 \cdot 10^{-3}(J/a)$. (e) Center of mass drift for the manually-loaded dark band (blue line) as in (b), the bare Hofstadter band (red line) as in (a), and for the realistically-prepared state (black crosses) as in (c); in the latter case, residual bright-state contributions were filtered out (main text), hence leading to the Chern number measurement of $C_D^{\mathrm{exp}} \approx 1.92$.

for the Chern number of the dark band $C_D^{\mathrm{exp}} = 1.99$ from the center-of-mass drift [Fig. 3(b)]. In contrast, the force is stronger and comparable to $\Delta_{\mathrm{db}}$ in the case depicted in Fig. 3(d), which leads to a dynamical repopulation of the bright states. Importantly, these undesired excitations are clearly identified by narrow stripes of highly populated sites, separated by $qa$ along the $x$ direction. For comparison, we also calculated the time-evolved density upon

Table 1: The percentage loss of the population due to the transfer to undesired states for different system sizes, detunings, and ramp durations.

| Detuning | final $A_s$ | Ramp duration $[\hbar/J]$ | $121 \times 11$ | $242 \times 22$ | $363 \times 33$ |
|---|---|---|---|---|---|
| $\delta_0 = J/50$ | 2 | $2 \cdot 10^3$ | | | 8.4% |
| $\delta_0 = J/50$ | 2 | $6 \cdot 10^3$ | | | 5.1% |
| $\delta_0 = J/50$ | 2 | $10^4$ | 7.2% | 3.6% | 3.2% |
| $\delta_0 = 0$ | 2 | $10^4$ | 8.1% | 5.2% | 5.3% |
| $\delta_0 = J/50$ | 0.5 | $10^4$ | | | 12.5% |

populating a bare Hofstadter band (with $C = 1$) instead of the dark band: the resulting Fig. 3(a), when compared with Fig. 3(b),(d), reveals that the center-of-mass velocity indeed differs by the predicted factor of two; see also Fig. 3(e).

Finally, we explore a realistic sequence for loading atoms into the designed dark band of Fig. 2. We start the sequence by completely filling the lowest Hofstadter band with a single species ($\sigma = 1$), in the absence of coupling to the other species ($\sigma = 2, 3$). This initial state is a Chern insulator with Chern number $C_1 = 1$. Considering the $\Lambda$ scheme of Eq. (13), one then progressively activates the couplings ($A_{1,2} \neq 0$) so as to adiabatically transfer the atoms from the bare Hofstadter band to the dark band with $C_D = 2$. Because of the change in the Chern number (which requires a gap-closing event), non-adiabatic effects are unavoidable, so that this final state cannot be reached with 100% fidelity. Moreover, the edge modes that cross the bulk gaps also provide a natural channel for non-adiabatic processes. For small systems, where the number of edge modes $O(L_x + L_y)$ is not marginal compared to the number of bulk modes $O(L_x L_y)$, we expect the dominant loss source to originate from the edge modes contribution.

An optimal population in the target dark band can be obtained by slowly ramping up the coupling $A_1$ to its final value [9, 73]. We perform a simulation of this sequence, by calculating the time evolution of single-particle states according to the time-dependent Hamiltonian $H_\Lambda[A_1(t)]$, where $A_1(t) = A_1(t/T)^\alpha$. In order to reduce the initial energy-degeneracy of three eigenstates at $t = 0$, we add a small and time-dependent detuning $\delta(t) = -\delta_0(1 - t/T)^{\alpha+1}$ to $\delta_1$, which indeed allows us to increase the efficiency of the process [73]. The particle density in the final state is shown in Fig. 3(c), where the thin vertical stripes separated by $qa$ again highlight the residual population in the bright states.

We illustrate the efficiency of the simulated loading scheme through Table 1. For instance, for a lattice of size $363 \times 33$, using a linear ramp of duration $T \approx 10^4 (\hbar/J)$ and a final value of the coupling strength $A_s = 2J$, we obtain a residual population fraction in the component $\sigma = 3$ of 3.2%; we remind that this population is a direct measure of the loss in our procedure. As visible in Table 1, the ramp duration is a crucial parameter: going from $2 \cdot 10^3 (\hbar/J)$ to $10^4 (\hbar/J)$ allows for a nearly threefold decrease of loss. One also observes that the detuning parameter $\delta_0$ introduced above significantly increases the loading efficiency (Table 1 compares $\delta_0 = J/50$ with $\delta_0 = 0$). Besides, we note that increasing the coupling strengths $A_{1,2}$ increases the dark to bright bands gap; accordingly, when considering the small value $A_{1,2} = 0.5J$, the losses significantly increase (12.5%) for the same lattice configuration.

We note that this undesired population can be significantly reduced by optimizing the ramp function (as in [74]), or by increasing the system size and ramp duration; other state-preparation protocols, based on optimal-control theory [75–77] and nonunitary dynamics [78], could also be designed to further maximize the fidelity. As far as the system size is concerned, we observe that the loading fidelity saturates as one increases the system size (close to $242 \times 22$ in our simulations); this indicates that the remaining losses are no longer dominated by transfer through the edge modes, but rather stem from the bulk gap closing during the ramp.

Finally, we calculate the time-evolution of the realistically prepared state under the action of a linear gradient along the $y$ direction, and show the resulting density in Fig. 3(d) for a drift duration $t = 10(\hbar/aF)$. After filtering out the bright-state contribution to the density, by removing the aforementioned stripes and restricting the measurement to well populated sites only, one obtains the center-of-mass motion depicted by crosses in Fig. 3(e), from which one extracts the "experimental" value $C_D^{\exp} = 1.92$. These simulations demonstrate that a dark band with $C_D = 2$ can be loaded and probed accurately using a realistic $\Lambda$ system [Eq. (13)].

The time-scales discussed in this work are expressed in terms of $\hbar/J$. The values of $J$ achieved in modern experiments correspond to several hundreds of Hz (e.g. 880 Hz$\times h$ in [79]). Considering such a value of $J$, the loading ramp duration discussed in the main text and in Fig. 3 would correspond to $T = 3 \cdot 10^3 \hbar/J = 540$ms. Besides, the total evolution time in the tilted lattice (Chern-number measurement) is $T = 2 \cdot 10^3 \hbar/J = 360$ms. Those durations are compatible with the coherence times of cold-atom experiments [79].

# 6 Conclusion

We proposed a realistic scheme by which flat bands with predictable higher Chern number can be constructed through a coherent manipulation of Bloch bands. While this work focused on the simplest $\Lambda$ configuration, more bands could be involved in the construction in view of building $N$-pod settings [80,81]; this strategy suggests a promising route to reach even higher Chern numbers $|C_D| \gg 2$. The dark Chern bands deriving from our scheme offer a platform where exotic fractional quantum Hall states could be explored with cold atoms, including generalized Moore-Read and Read-Rezayi states [24, 26], topological nematic states [23] and "genon" defects [23, 29, 82]. Interesting perspectives concern the fate of dark Chern bands in the presence of dissipation, and the coupling of Bloch bands belonging to other topological classes [83].

# Acknowledgements

The authors gratefully acknowledge A. Bochniak, B. Mera and A. Sitarz for insightful discussions, which culminated in the proof of the sum rule in Eq. (9). They also thank C. Repellin for her thoughtful comments on the manuscript, and for highlighting the possibility of realizing genons in this higher-Chern-number context. They finally thank B. Irsigler for pointing out the existence of dark states and sum rules for Chern numbers in multilayer systems [63], and M. Lewenstein for triggering this collaboration.

**Funding information** Support of the National Science Centre (Poland) via grants 2016/23/D/ST2/00721 (M.Ł) and 2016/21/B/ST2/01086 (J.Z.) is acknowledged. N.G. is supported by the ERC Starting Grant TopoCold, and the Fonds De La Recherche Scientifique (FRS-FNRS, Belgium). This research was supported in part by PLGrid Infrastructure.

## A  Dark state formula

The dark state formula [Eq. (3)] for the three level $\Lambda$ system in Eq. (2) is valid in the case where $\delta = 0$; no simple analytical formula for the dark state is known for the general Hamiltonian in Eq. (2). As moderate values of $\delta, \Delta$ are used in practice, we can use $\delta/\Omega_{1,2}$ and $\Delta/\Omega_{1,2}$ as small parameters from which a perturbative expansion can be constructed. For finite $\delta$, admixtures lead to corrections to the dark state's energy,

$$E_0 \approx \frac{\delta |\Omega_1|^2}{|\Omega_1|^2 + |\Omega_2|^2},\tag{15}$$

and to the state itself,

$$|D(\delta)\rangle = |D(0)\rangle + \delta \frac{\Omega_1 \Omega_2}{(|\Omega_1|^2 + |\Omega_2|^2)^{3/2}} |3\rangle \ldots,\tag{16}$$

and it is then referred to as the "gray" state. The above equation reveals the lowest-order correction to the dark state, which only involves the excited state $|3\rangle$. Next terms that are $O(\delta^2)$ and $O(\delta\Delta)$ involve all three base states $|1\rangle, |2\rangle, |3\rangle$.

Importantly, when setting $\delta = 0$, the detuning $\Delta$ does not affect the $E_0$ eigenvalue and the state $|D(0)\rangle$ remains to be an accurate dark state. As a result, in a $\Lambda$ scheme made of Bloch bands [Fig. 1(b) in the main text], the flatness requirements on the band associated with the "third" state $|\psi_3(\mathbf{k})\rangle \equiv |3\rangle$ are substantially relaxed with respect to the other two bands.

In the standard situation where $\Omega_1$ and $\Omega_2$ couple different atomic states, the excited state $|3\rangle$ undergoes spontaneous emission. However, the spontaneous emission rate (typically in MHz range) dominates over single kHz energy scales for ultracold atom dynamics, and consequently even a small admixture to the dark state is detrimental.

In the setting considered in this work, where the excited state $|3\rangle$ is chosen as a stable optical lattice band, the admixture implied by (16) does not imply larger losses, and the Chern number is invariant with respect to perturbations of the topological dark band. This holds as long as the band gap separating dark and bright bands does not close. In fact, as seen in the following section, this requires $\delta/\Omega_{1,2} \ll 1$.

## B  Rotating Wave Approximation

We now consider three Bloch states with fixed quasimomentum $\mathbf{k}$ that are coupled by two time-dependent processes of frequencies $\omega_{1,2}$. We hereby discuss the validity of Eq. (1) in the main text in describing this setting.

We consider three states $|\psi_1(\mathbf{k})\rangle, |\psi_2(\mathbf{k})\rangle, |\psi_3(\mathbf{k})\rangle$ with energies $E_{1,2,3}$. The Hamiltonian is

written as

$$H_{\Lambda,\text{lab}} = \hbar \underbrace{\begin{pmatrix} E_1 & 0 & 0 \\ 0 & E_2 & 0 \\ 0 & 0 & E_3 \end{pmatrix}}_{H_{at,lab}} + A\sin(\omega_1 t) + B\sin(\omega_2 t), \tag{17}$$

where $A, B$ are operators that describe the couplings,

$$A = \hbar \begin{pmatrix} A_{11} & A_{12} & A_{13} \\ A_{12}^* & A_{22} & A_{23} \\ A_{13}^* & A_{23}^* & A_{33} \end{pmatrix}, \tag{18}$$

and similarly for $B$. We now transform to the rotating frame with the transformation $\psi = U^\dagger \psi_{\text{lab}}$ where $U^\dagger = \text{diag}[\exp(iE_1 t/\hbar), \exp(i(E_1 + \hbar\omega_1 - \hbar\omega_2)t/\hbar), \exp(i(E_1 + \hbar\omega_1)t/\hbar)]$. The transformation to the co-rotating frame is achieved by: $H_\Lambda = U^\dagger H_{\Lambda,\text{lab}} U - i\hbar U^\dagger \dot{U}$, where the dot denotes the time derivative. Specifically $H_{at,rot}$ transforms as $H'_{at,rot} = \text{diag}(0, \delta, -\Delta)$, where $\delta = E_2 - E_1 - \hbar\omega_1 + \hbar\omega_2$ and $\Delta = E_1 - E_3 + \hbar\omega_1$; note that it includes the terms from the diagonal operator $-i\hbar U^\dagger \dot{U}$. The remaining part of the Hamiltonian transforms as $A' = U^\dagger A U \sin(\omega_1 t)$ and $B' = U^\dagger B U \sin(\omega_2 t)$, namely

$$A' = \frac{\hbar}{2i} \begin{pmatrix} -2iA_{11}\sin(\omega_1 t) & A_{12}e^{i\omega_2 t}(e^{-2i\omega_1 t} - 1) & A_{13}(e^{-2i\omega_1 t} - 1) \\ A_{12}^* e^{-i\omega_2 t}(1 - e^{2i\omega_1 t}) & -2iA_{22}\sin(\omega_1 t) & -2iA_{23}e^{-i\omega_2 t}\sin(\omega_1 t) \\ A_{13}^*(1 - e^{2i\omega_1 t}) & -2iA_{23}^* e^{i\omega_2 t}\sin(\omega_1 t) & -2iA_{33}\sin(\omega_1 t) \end{pmatrix},$$

and analogously for $B'$.

In the situation where all the states $|\psi_i(\mathbf{k})\rangle$ belong to different spin manifolds, the Rotating Wave Approximation (RWA) directly applies and all the time-dependent, rapidly oscillating terms in the above matrix $A'$ may be neglected. One then obtains

$$A' \approx \frac{\hbar}{2i} \begin{pmatrix} 0 & 0 & -A_{13} \\ 0 & 0 & 0 \\ A_{13}^* & 0 & 0 \end{pmatrix}. \tag{19}$$

If two states $|1\rangle$ and $|2\rangle$ are taken within the same spin manifold, the above approximation should be replaced with

$$A' \approx \frac{\hbar}{2i} \begin{pmatrix} 0 & 0 & -A_{13} \\ 0 & 0 & -A_{23}e^{i(\omega_1 - \omega_2)t} \\ A_{13}^* & A_{23}^* e^{-i(\omega_1 - \omega_2)t} & 0 \end{pmatrix}.$$

Indeed, the time-dependent terms $\pm A_{23}^{(*)} e^{\mp i(\omega_1 - \omega_2)t}$ contain the frequency $(\omega_1 - \omega_2)$, which is orders of magnitude smaller then all others frequencies. In fact, it is typically of the order of $J/\hbar$, as established by the Bloch band's bandwidth. However, we point out that the approximation in Eq. (19) is still justified in this case, as long as the amplitude verifies $A_{23} \ll J$; this approximation was assumed to be valid in the main text, when describing the $\Lambda$ system by Eq. (2). For instance, when $A$ is generated by a lattice modulation, the coefficients $A_{23}, A_{23}^*$ may be minimized by simply lowering the amplitude of the modulation. Alternatively, when different spin states are involved, $A_{23}$ can vanish due to selection rules. The justification for neglecting the terms oscillating with frequency $(\omega_1 - \omega_2)$ is much more

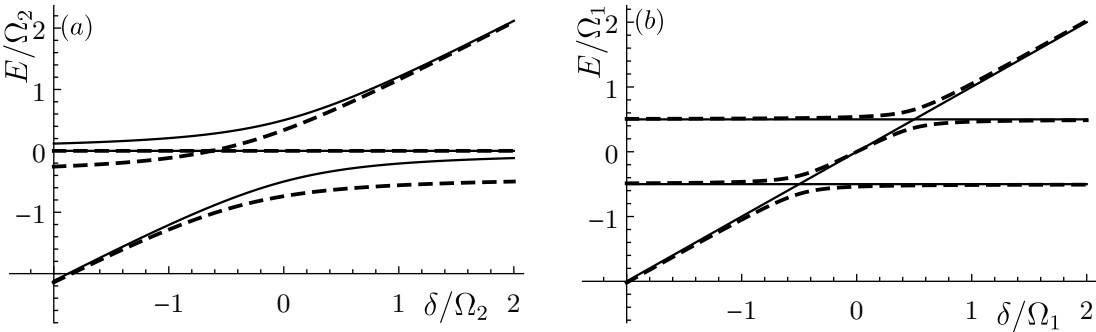

Figure 4: Three energy levels of the $H_\Lambda$ Hamiltonian for special values of $\Omega_1(\mathbf{k}), \Omega_2(\mathbf{k})$ as a function of detuning from Raman resonance condition $\delta$. Panel (a) treats the case $\Omega_1(\mathbf{k}) = 0, \Delta(\mathbf{k}) = 0$ (solid lines), $\Omega_1(\mathbf{k}) = 0, \Delta(\mathbf{k}) = 0.4\Omega_2(\mathbf{k})$ (dashed lines). Panel (b) shows the case $\Omega_2(\mathbf{k}) = 0, \Delta(\mathbf{k}) = 0$ (solid line), $\Omega_2(\mathbf{k}) = 0.4\Omega_1(\mathbf{k}), \Delta(\mathbf{k}) = 0$ (dashed line).

evident when the states $|1\rangle$ and $|2\rangle$ correspond to different hyperfine states; then $\hbar(\omega_1 - \omega_2)$ is of the order of the hyperfine splitting, which is several orders of magnitudes larger than $J$.

The same reasoning applies to the operator $B$, with $B_{12}$ and $B_{23}$ replacing the coefficients $A_{12}$ and $A_{23}$.

## C   Dark state stability and the flatness of bands

### C.1   Dark state stability

The stability of the dark state is understood as the existence of a finite gap to the remaining two bright eigenstates in Eq. (2). Here we analyze the effects of finite $\delta$ on these gaps.

For $\delta(\mathbf{k}) = 0$ the bright state eigenenergies are:

$$E_\pm = \frac{1}{2}(-\Delta(\mathbf{k}) \pm \sqrt{\Delta^2(\mathbf{k}) + 4\bar{\Omega}^2(\mathbf{k})}). \tag{20}$$

Since the dark state energy is $E_0 = 0$, $|E_\pm(\mathbf{k})|$ also gives the amplitude of the gap between bright and dark states. It is simple to check numerically that if both $\Omega_s(\mathbf{k}) \neq 0$, then the gap remains finite (though possibly small) for certain values of $\delta(\mathbf{k})$ or $\Delta(\mathbf{k})$.

When exactly one of the Rabi couplings ($\Omega_1(\mathbf{k})$ or $\Omega_2(\mathbf{k})$) is zero, one numerically obtains that the gap may close. The other, non-zero Rabi frequency can then be chosen as the energy unit.

We first consider the case when $\Omega_1(\mathbf{k}) = 0$ and $\Omega_2(\mathbf{k}) \neq 0$. Fig. 4(a) shows the eigenvalues of $H_\Lambda$ as solid, thin lines in that case. For $\delta(\mathbf{k}) = 0$, as expected, the dark state energy is precisely 0. We see that for large $|\delta(\mathbf{k})|$ the energy of one of the bright states become close to the energy of the dark/gray state, though never crosses it. The gap value is, therefore, substantially lowered which may lead to the depletion of the dark/gray state in the experiment. For $\Delta(\mathbf{k}) \neq 0$ the dark/gray energy level is crossed by [see Fig. (4)(a)] the bright state at a finite value of $\delta(\mathbf{k})$ (dashed lines). For $\Delta(\mathbf{k}) \neq 0$ the crossing occurs for finite value of $\delta(\mathbf{k})$.

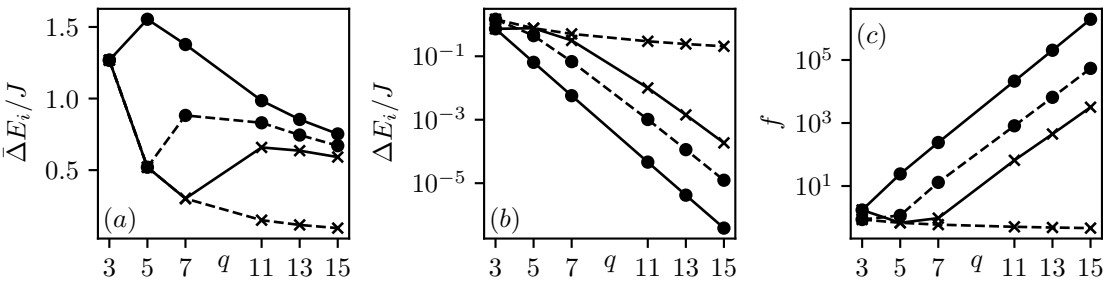

Figure 5: The panels $(a), (b)$ and $(c)$ show the properties of bands $b = 1, 2, 3, (q-1)/2$ of HH $(H_{HH})$ model with $\phi = 2\pi/q$. The bands are marked by ●—●, ●··●, ✕—✕, and ✕··✕ respectively. Panel $(a)$ shows the bandgap $\bar{\Delta} E_i$ to neighboring band, and Panel $(b)$ — the bandwidth $\Delta E_i$. Panel $(c)$ shows the flatness ratio $f$ for the same bands.

For example for $\Delta(\mathbf{k}) \approx 0.4\Omega_2(\mathbf{k})$, the crossing takes place when $\delta(\mathbf{k})/\Omega_2(\mathbf{k}) \approx -0.6$, and the smaller the $|\Delta(\mathbf{k})|$, the larger $\delta(\mathbf{k})$ is necessary for the crossing to occur.

When $\Omega_2(\mathbf{k}) = 0$ and $\Omega_1(\mathbf{k}) \neq 0$, the dark/gray state energy line is exactly linear with $\delta(\mathbf{k})$ as $|D(\mathbf{k})\rangle = |\psi_1\rangle$ and $|D(\mathbf{k})\rangle$ is decoupled from $|\psi_2(\mathbf{k})\rangle$ and $|\psi_3(\mathbf{k})\rangle$. In particular, for $\delta(\mathbf{k}) = \pm\Omega_2(\mathbf{k})/2$ the gap to the bright state closes. For $\Omega_2(\mathbf{k}) \neq 0$ this energy level crossing becomes avoided [see Fig. 4(b)]. In this case the value of the detuning $\Delta(\mathbf{k})$ does not change the location of the crossing, or whether it is avoided or not.

The presence of the energy level crossings (either exact or narrowly avoided) would have a detrimental effect on the construction of the dark-state band with a well-defined Chern number. However, in both cases discussed above, the crossings require large values of $\delta(\mathbf{k})$: i) $\Omega_1(\mathbf{k}) = 0$ and $\delta(\mathbf{k}) = O(\Omega_2(\mathbf{k}))$ or ii) $\Omega_2(\mathbf{k}) = 0$ and $\delta(\mathbf{k}) = O(\Omega_1(\mathbf{k}))$. Nevertheless, when the dependence of $\Omega_{1,2}(\mathbf{k})$ on the quasimomentum $\mathbf{k}$ is considered, one notices that $\Omega_{1,2}(\mathbf{k})$ may have zeros where $\Omega_{1,2}(\mathbf{k})$ is also small relative to its maximum value. When the values of $\Omega_{1,2}(\mathbf{k})$ rapidly oscillate across the BZ, a strong upper bound on $\delta(\mathbf{k})$ is therefore implied.

## C.2   Band flatness and Rabi frequencies variation in the Harper-Hofstadter model

The dark state formula, Eq. (3) and Eq. (16), accurately describe the states in the dark Chern band under the following assumptions:

1. For each value of quasi-momentum $\mathbf{k}$, the detunings $\delta(\mathbf{k})$ due to the finite bandwidth of the bands supporting states $|\psi_1(\mathbf{k})\rangle \equiv |1\rangle, |\psi_2(\mathbf{k})\rangle \equiv |2\rangle, |\psi_3(\mathbf{k})\rangle \equiv |3\rangle$ are small compared to $\Omega_1(\mathbf{k}), \Omega_2(\mathbf{k})$ (as discussed in Section C).

2. The detuning $\Delta(\mathbf{k})$ should not be too large, as the gap separating dark and bright bands approaches $\bar{\Omega}^2(\mathbf{k})/4\Delta(\mathbf{k}) \to 0$ when $\Delta(\mathbf{k}) \gg \bar{\Omega}(\mathbf{k})$.

3. States $|\psi_1(\mathbf{k})\rangle, |\psi_2(\mathbf{k})\rangle, |\psi_3(\mathbf{k})\rangle$ are decoupled from any other states (other bands, other atomic states).

The first two requirements can be summarized in the following inequality (for each quasi-momentum $\mathbf{k}$):

$$\delta(\mathbf{k}) \ll |\bar{\Omega}(\mathbf{k})| \ll \bar{\Delta} E_i, \quad i \in \{1, 2, 3\}. \tag{21}$$

Here $\bar{\Delta}E_i$ is the minimum distance of the band associated with bare state $|i\rangle$ to its nearest neighbors. In the example discussed in the main text, $\bar{\Delta}E_1 = O(J)$ simply corresponds to a gap between the lowest two Hofstadter bands; the gap $\bar{\Delta}E_2$ is between the top and second-top bands [in the specific choice of bands described in the main text, $\bar{\Delta}E_2 = \bar{\Delta}E_1$ due to the symmetry of the spectrum; see Fig. 2(a)]; the gap $\bar{\Delta}E_3$ is the standard separation between $s$ and $p$ bands, in an optical lattice potential $V(x,y) = V_x \cos^2(kx/q) + V_y \cos^2(ky)$. The latter gap is $\bar{\Delta}E_3 \approx \min\{\sqrt{4E_R V_y}, \sqrt{4E_R V_x/q^2}\}$, where $E_R$ is the recoil energy; sufficiently large optical-potential amplitudes $V_x, V_y$ can easily ensure sufficiently large $\bar{\Delta}E_3$. Altogether, only $\bar{\Delta}E_{1,2}$ give an upper bound on practical Rabi frequencies $\Omega_{1,2}(\mathbf{k})$.

It is useful to consider the ratio measuring the total variation of the Rabi frequency across the BZ:

$$g = \min_{\mathbf{k} \in BZ} |\bar{\Omega}(\mathbf{k})| / \max_{\mathbf{k} \in BZ} |\bar{\Omega}(\mathbf{k})|. \tag{22}$$

The value of this quantity provides an indication on how flat the bands associated with the states $|\psi_1(\mathbf{k})\rangle$ and $|\psi_2(\mathbf{k})\rangle$ have to be in view of tuning the overall amplitude of the Rabi frequencies so as to satisfy inequality (21). The flatness of the state $|\psi_3(\mathbf{k})\rangle$ is not so crucial [see Appendix A].

The definition of $g$ depends implicitly on the choice of the bands $b_{1,2}$; we will use the subscripted notation $g_{b_1 b_2}$ below, where $b_1, b_2$ indicate the bands chosen from the Hofstadter spectrum for the states $|1\rangle, |2\rangle$ in the $\Lambda$ system.

For a particular choice of the model and bands, we consider the following flatness $f$-factor:

$$f = \bar{\Delta}E_i / \Delta E_i, \tag{23}$$

where $\Delta E_i$ indicates the bandwidth of a given band. The inequality (21) can be satisfied when

$$f \gg g^{-1}. \tag{24}$$

The following two subsections will discuss the values of $f$ and $g$ in different cases, with an emphasis on cases where the former is maximized and the latter is minimized.

## C.3   Band flatness — flatness factor $f$

We consider the Hofstadter-Harper model under PBC in the thermodynamical limit [for numerics, system sizes $O(100 \times 100)$ suffice]. In Fig. 5(a), (b) and (c) we show the bandgap $\bar{\Delta}E_i$, bandwidth $\Delta E_i$ and the flatness ratio $f$ for different $q$, for the lowest three bands and the central band (with $C = -q + 1$) of the HH model. We notice that $\bar{\Delta}E_i/J$ is always $O(1)$, except for the band with $C \neq 1$ where it quickly drops, and that the bandwidth $\Delta E_i/J$ drops exponentially with $q$ already for the considered moderate values of $q$. As a result, the factor $f$ exponentially increases with $q$ for bands with $C = 1$.

The behavior for $C = -q+1$ is starkly different. The bandgap decreases as $\sim 1/q^2$ and the bandwidth also decreases like $\sim 1/q$ with $q$. In the end, the maximal flatness factor $f \approx 6.5$ for the middle band is for $q = 6$ and it drops down to zero as $\sim 1/q$.

In light of this analysis, the use of the middle band (with $C \neq 1$), to provide states $|1\rangle$ or $|2\rangle$ in our $\Lambda$ system, is questionable.

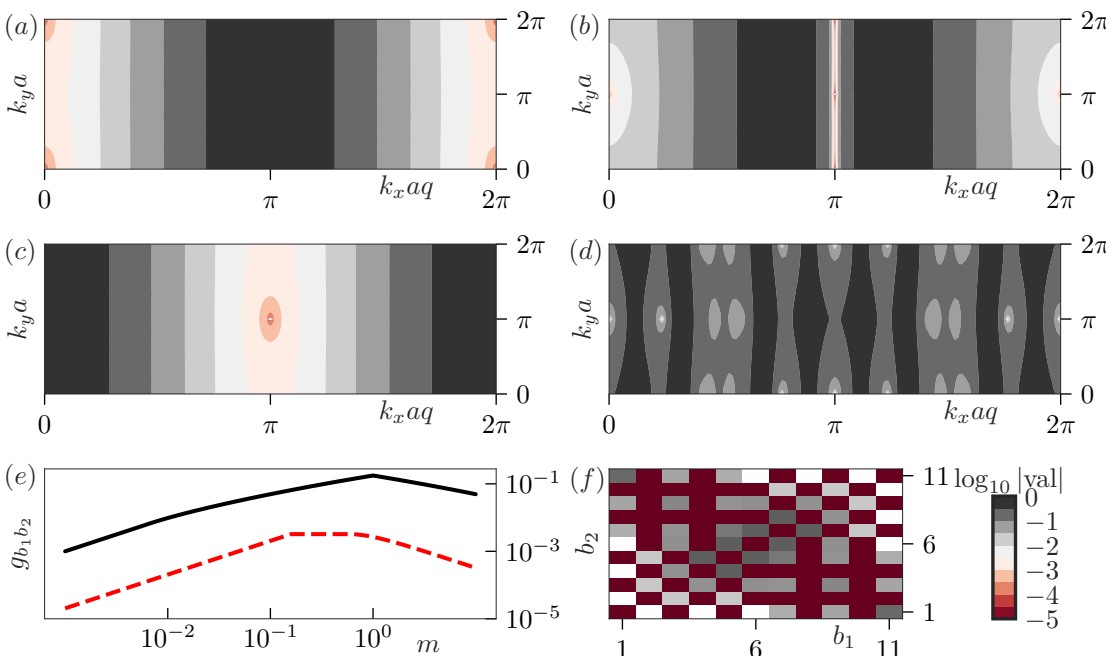

Figure 6: Panels $(a - d)$ show **k**-dependence across the entire BZ of $\log_{10}|\Omega(\mathbf{k})|$, the Rabi frequency coupling a trivial flat $s$-band to lowest, first-excited, top, and middle band of a HH model, for a flux $\phi = 1/11$. Here, $\max_{\mathbf{k}}|\Omega(\mathbf{k})| = 1$ sets the normalization. Panel $(e)$ shows factor $g_{1,11}$ $(g_{1,2})$ as a function of $m = \max_{\mathbf{k}\in BZ}|\Omega_1(\mathbf{k})|$ with solid, black (dashed red) lines. Panel $(f)$ shows $\log_{10} g_{b_1 b_2}$ for $m$ maximizing its value for different $(b_1, b_2)$. Red squares correspond to $(b_1, b_2)$ where $g_{b_1 b_2} = 0$.

## C.4 Rabi frequencies total variation — factor $g$

Here we detail the dependence of the Rabi frequencies $\Omega_{1,2}$ on the quasimomentum **k** for the HH model. This aspect is important in view of satisfying Eq. (21).

First let us remark that the Rabi frequency coupling two bands with different Chern number has to be zero somewhere in the BZ. Indeed, if we assume *a contrario*, that $\Omega_s(\mathbf{k}) \neq 0$, for all $\mathbf{k} \in$ BZ, we can construct the following system: $H = \delta|\psi_s\rangle\langle\psi_s| + \Omega_s(\mathbf{k})\left(|\psi_s\rangle\langle\psi_3| + |\psi_3\rangle\langle\psi_s|\right)$ for all $\mathbf{k} \in BZ$. If the detuning $\delta$ is swept from a large positive value $\delta \gg |\Omega_s(\mathbf{k})|$ to a large, negative value: $\delta \ll -|\Omega_s(\mathbf{k})|$ then the eigenstate of this model changes adiabatically between $|\psi_3\rangle$ and $|\psi_s\rangle$, which would imply $C_3 = C_s$, violating our assumptions. This argument says nothing about $\Omega_s(\mathbf{k})$ connecting bands with same Chern number.

We numerically find that in the example studied in the main text, where the two bands $b_1$ and $b_2$ come from the HH model, and the band $b_3$ is a $s$ band of a (topologically trivial) 2D optical potential, $\Omega_1(\mathbf{k})$ and $\Omega_2(\mathbf{k})$ have coinciding zeros for roughly half of the choices of the two Hofstadter bands [see Fig. 6(f)] .

In Fig. 6(a)-(d) we plot the coupling strength $\log_{10}|\Omega|$ between the $i$-th band of the HH model [for $i = 1, 2, q, (q+1)/2$ respectively] and a trivial flat $s$-band. We remind that the Rabi frequencies $\Omega_s(\mathbf{k})$ are due to terms proportional to $A_s$ [in Eq. (6)]. The $|\Omega_s|$ attains a single zero at $(k_x, k_y) = (0, 0)$ for coupling to the first band, at $(k_x, k_y) = (\pi, \pi), (\pi, 0), (0, \pi)$ for coupling to the second band, and at $(k_x, k_y) = (\pi, \pi)$ for coupling to the highest band.

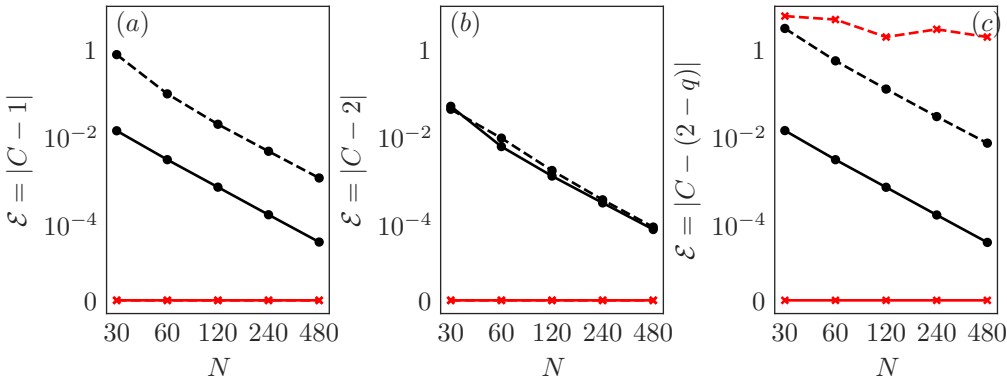

Figure 7: Approximation error $\mathcal{E}$ in Chern number computation. Panel $(a)$ shows the residual in Chern number, $C = 1$ computation of lowest band of simple HH model. Panels $(b)$, $(c)$ show same for dark state Chern number. In $(b)$: $C_D = 2$ for $b_1 = 1$ and $b_2 = q$. and for panel $(c)$: $C_D = -q + 2$ for $b_1 = 1$ and $b_2 = (q+1)/2$. The curve markings are common and refer to $\bullet\!\!-\!\!\bullet$, brute-force discretization method, $q = 3$, $\bullet\!\cdot\!\cdot\!\bullet$ brute-force discretization method, $q = 11$, $\times\!\!-\!\!\times$ Fukui method, $q = 3$ and $\times\!\cdot\!\cdot\!\times$ Fukui method, $q = 11$.

The value of $g_{b_1 b_2}$ does not change if both $\Omega_1(\mathbf{k})$ and $\Omega_2(\mathbf{k})$ are multiplied by a common factor. We assume $\max_{\mathbf{k}} |\Omega_2(\mathbf{k})| = 1$ and $\max_{\mathbf{k}} |\Omega_1(\mathbf{k})| = m$. Figure $6(e)$ shows $g_{1,11}$ as a function of $m$ [compare panels $(a)$ and $(c)$]. For some $m = m_*$ the $g_{b_1 b_2}$ attains the maximum value. It is also evident that the band choice can alter the factor $g_{b_1 b_2}$ by several orders of magnitude. For the choice of $(b_1, b_2) = (1, 11)$ for $q = 11$ the maximal value of $g_{1,11}$ is $g_{1,11} \approx 0.3$.

Figure 6 $(f)$ shows $\log_{10} |g_{b_1 b_2}|$ for different choice of $b_1, b_2$ bands as a color array plot (always for $m = m_*$). We notice that for some values (indicated by deep red square) of $b_1, b_2$ (for example $b_1 = b_2$ or $b_1 = 1, b_2 = 3$) we have $g_{b_1 b_2} = 0$. This is due to coinciding zeros of $\Omega_1(\mathbf{k})$ and $\Omega_2(\mathbf{k})$.

## C.5 Numerical evaluation of Chern numbers

The Chern number is defined as the integral of the Berry curvature over the BZ [Eq. (6)]. The integral can be directly computed using standard methods for numerical integration. In this work we used a regular discretization of BZ into $N \times N$ evenly-sized pieces, and integration using trapezoid prescription. When $N$ is sufficiently large, the integral for $C$ gives an error $\mathcal{E} \sim N^{-2}$ as expected. In particular, applying this method, the approximation for $C$ is not an integer number.

An alternate method has been proposed by Fukui et. al. [84]. There the integration is performed also by summation over rectangular plaquettes, but the approximant for the Chern number integral is manifestly gauge invariant, and the Chern number approximation is guaranteed to be an integer.

Figure 7(a) shows the comparison of the two methods. There we have computed the Chern numbers of the lowest band ($C = 1$) for the standard HH model for $q \in \{3, 11\}$ using brute-force discretization and Fukui's method. In this case the method proposed by Fukui offers an obvious numerical advantage, and returns correct Chern numbers even for smallest

considered discretizations. In Panel ($b$) of the same Figure we compute the Chern number of the dark state ($C_D = 2$) as discussed in the main text. Panel ($c$) shows the computation of the Chern number of a dark state band where $|1\rangle, |2\rangle$ correspond to the lowest and middle bands of the HH model ($C_D = -q + 2$). In the latter case the brute force approach converges up to the correct result much faster than the Fukui's method.

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
