# Peer review of "A dark state of Chern bands: Designing flat bands with higher Chern number"

_SciPost Physics_

## Round 3 · Referee Report · Anonymous (Referee 1) · 2020-10-21

Strengths

1) The article proposes a way of producing novel topological band structures that have a flat band characterised by a non-zero integer Chern number. The authors present a particular realisation of a flat band with a Chern number equal to 2, and describe how one could generalise this to arbitrary integers. 2) This proposal is feasible and sensible with an experimental realisation with ultra-cold atoms in optical lattices. 3) The authors provide proof-of-principle simulations for detecting the topological Chern number in current cold atom experiments. 4) This work is novel and interesting and provides a way of preparing and detecting novel topological features directly in an experimental setting. This has the potential to have a high impact in the field.

Weaknesses

1) When the authors discuss experimental preparation and detection, specifically the timescales that are required experimentally, they do not discuss the experimental limitations. For example, some of the timescales that they quote in section 5 – for typical experimental parameters – would correspond to many seconds or even minutes, which are out with the capabilities of current experiments due to decoherence effects. It is a little unclear how their conclusions would change if more realistic values were used in their simulations.

Report

This is an interesting and well written article. The results are presented clearly, the experimental proposals are for the most part sensible and the authors conclusions are backed up by numerical simulations.
The ability to experimentally prepare novel topological band structures opens up new avenues for exploring the behaviour of topological systems. In particular, being able to realise these systems in experiments with ultra-cold atoms offers a feasible way of including interactions between atoms allowing for investigations into interacting topological systems such as fractional quantum hall phases.

In this case I would recommend publication in this journal. However, I suggest that the authors address the below points, as I believe that this will increase the impact of their article.

Requested changes

1) In the figure 3 caption, the authors state that for (a) they use a value of $Fa=5\times 10^{-4} J$ and a timescale of $t=15 \hbar/J$. In typical cold atom experiments, $J$ usually takes values around $J/h \approx 100 \rightarrow 1000 ~{\rm Hz}$. If the larger values are used then this will then correspond to a timescale of $t\approx 5~{\rm s}$. These values for the timescales seem large compared to typical experimental coherence times of $< 1 {\rm s}$. Can the authors discuss how important these timescales are for detecting the Chern number in the way that they are proposing. For (b) they use a value of $Fa=5\times 10^{-3} J$, which then corresponds to $t\approx 0.5~{\rm s}$ - which seems more experimentally feasible. But if smaller tunnelling amplitudes are realised (see Ref.[9] in the article for example) then even the timescales here may be too large.

2) Similarly, on page 9, when discussing preparing atoms in the flat band, they quote timescales for their linear adiabatic ramp of $T=10^6 \hbar/J = 160~{\rm s}$. The authors do state that this can be improved for different ramping procedures, but they do not present specific simulations or estimates. Can the authors discuss the errors in this preparation scheme for more experimentally realistic timescales, such as $T<1~{\rm s}$.

---

## Round 3 · Referee Report · Anonymous (Referee 2) · 2020-11-19

Strengths

1) A new idea to generate a topologically nontrivial dark band in cold atom systems is introduced. 2) A simple sum rule is presented substantiated by detailed numerical simulations. 3) Experimental protocols to verify their findings are clearly demonstrated.

Weaknesses

1) Many important details are left in Appendices, which could be integrated into the main texts for better readability.

Report

This work introduces an interesting idea to generate a topologically nontrivial dark band in cold atom systems. The sum rule of Chern numbers is unique and extensively verified by the numerical simulations. In particular, the authors proposed a setup to generate $C_D = 2$ band, which is shown to be experimentally detectable by the center-of-mass responses.

The findings are important for further investigation of exotic topological phases in cold atom systems, and this work meets the criteria for publication. Therefore, I recommend the publication of the manuscript.

I appreciate it if the authors can clarify the following points to improve the presentation.

Requested changes

In Sec. 4, a new setup is introduced to generate a $C_D = 2$ band. I have several questions on this:

1) While it is understandable that Eq. (6) produces one dark band and two bright bands, its connection to the previous $\Lambda$ system is not clear. Here all the three bands are overlapping without the coupling $A_s$, thus it doesn't look like a $\Lambda$-type. 2) What is the value of $A_s$ in Fig. 2(c), and its relation to $\Omega_s(\mathbf{k})$ (in the caption of Fig. 2)? 3) Finally, why is the trivial band so flat? If naively $\hat{H}_\text{HH}$ with $\phi = 0$ is taken, the bandwidth is $\sim J$. Is it due to the fact that only every $q$ sites are used along the $x$-axis, which implies $J_x \ll J_y$?

---

## Round 3 · Referee Report · Anonymous (Referee 3) · 2020-11-20

Strengths

-new and interesting result
-simulation of the center-of-mass response relevant to the experimental realization

Weaknesses

-the comparison of the related works should be improved
-very little intuition behind the main result of CD=C1+C2-C3
-numerics justifying the expression for CD is lacking--it is only stated in the appendix that the authors checked the expression for CD in a number of scenarios. Given the lack of analytical results and intuition, this should be improved by presenting their numerics explicitly; and probably also showing when this expression stops to work (when?).

Report

The authors propose how to engineer the high Chern number bands by coupling multiple topologically nontrivial bands using external light fields. They discuss how to prepare and measure the system in the state of interest.

The work seems to be very similar to Panas2020. There, three levels are studied, the dark state considered, but the coupling is different.
-The discussion relating the two should be extended.
-What are the strengths of this approach compared with Panas2020? What are the weaknesses? How does it compare with other approaches to (flexibly?) engineering C>1?
-Why CD=C1+C2-C3 here, but there is simply CD=C1+C2+C3? What makes the difference?
-If there is no intuitive explanation, could the authors explain why it is hard to come up with an explanation? It seems that Panas2020 hasn't given too much intuition as well, but could one cook up a simple toy model to illustrate the physics?
-Dark state has a negligible contribution from |3> state, so why does it contribute to the Chern number.
-Why the phase between states 0 and 1 does not contribute (or does it?) to the expression of CD?
-What if the light coupling between the bands has an angular momentum? Would it lead to different results?
-Why do we need three levels? Is dark state considered so that we have flatter bands? Or we want to simply avoid lossy state |3> and still have strong and tunable coupling between |1> and |2>. Could authors shed more light on that?
-What if we consider coupling between two long-lived bands: do we have C=C1\pm C2 for some circumstances? Which sign should it be?
-For the M-scheme with states 1-2-3-4-5 (the dash indicates which re coupled) with the dark state being a superposition of 1,3, and 5, what is the expression for CD? Is it CD=C1+C3+C5-C2-C4?

-I agree with Referee 2 when it comes to the experimental parameters.

I believe that the work is interesting and worth publishing in the end in some journal. Before however making my final decision, I think that presentation could be improved for the profit of a reader.

Minor:
-what makes HH bands reminiscent of the Landau levels?

Requested changes

Please, address my question in the Report above.

Minor:
Though the draft is in general well written the Authors could make sure that there are no more shortcoming similar to which I bought from my quick read:
-Eq 4, why not add subscript \nu to F in Eq 4, I think it would make the notation clearer.
-Abstract:
--remove comma in 'band", by coupling'
--replace 'very flat' by e.g. 'nearly flat'
-remove comma in 'approach, in view'

---

## Round 4 · Referee Report · Anonymous · 2021-4-12

Report

With reference to my previous report, my main criticisms have been addressed. The authors have provided clarification on the timescales required for preparing and detecting the proposed features, indicating that they are in principle accessible experimentally.

In this case I would recommend publication in this journal.

---

## Round 4 · Referee Report · Anonymous · 2021-4-19

Report

I appreciate the changes and efforts made by the authors that have improved the quality of the manuscript. The issues that I raised in the previous comments have been cleared to my satisfaction. Therefore, I recommend the publication of the manuscript.

---

## Round 4 · Author Response

Dear Editors,

We hereby resubmit our manuscript entitled "A dark state of Chern bands: Designing flat bands with higher Chern number".

We appreciated the very constructive and thoughtful reports of the three Referees, which we hereby acknowledge. A reasonable list of questions and suggestions for improvements were formulated in these reports, which invited us to perform a substantial revision of our work.

Most importantly, our revised manuscript now contains: (a) a mathematical proof of the sum rule [Eq. 9], which constitutes the central result of our work; (b) new numerical simulations of the loading and detection schemes, which are now compatible with realistic experimental time scales; (c) a clearer and more pedagogical description of the Lambda system.

A point-by-point answer to the Referees' comments are provided below, as well as a list of changes.

We believe that the revised manuscript fully addresses all the concerns and suggestions of the Referees, making it suitable for publication.

N. Goldman, on behalf of the authors.

%%%%%%%%%%%%%%%%%%%%%%%%%%%

Reply to Referee 1:

We thank Referee 1 for their careful reading of our work. We are very pleased that the Referee finds our work interesting and suitable for publication. We are also very grateful for their insightful remarks regarding experimental aspects.

The requested changes concerned the timescales of the proposed protocol in view of experimental implementation. We have thoughtfully addressed this issue in the revised manuscript, which presents new simulations of the loading protocol and Chern-number measurement. In particular, Table 1 now explicitly presents the error (transfer to the undesired states) for various system sizes, ramp durations and other system parameters. The main results shown in Fig. 3 now also correspond to more realistic time scales (<1s), compatible with experimental coherence times. We note that a new ramp profile has been designed in view of optimizing the protocol (allowing for ramp durations of the order of 10^3 hbar/J); we also improved the duration of the Chern-number measurement (now also of the order of 10^3 hbar/J) by increasing the strength of the Rabi couplings. Under these conditions, the total Hall drift corresponds to about 30 lattice sites, which is satisfactory in view of detection.

We believe that our revised manuscript takes all the suggestions of the Referee into account, and that it is now in a suitable form for publication.

%%%%%%%%%%%%%%%%%%%%%%%%%%%

Reply to Referee 2:

We thank Referee 2 for their careful reading of our manuscript. We are pleased that the Referee finds our work interesting and suitable for publication.

Following a suggestion of the Referee, we now moved some informations from the Appendix to the main text. In particular, this should clarify some important notions and definitions inherent to our \Lambda scheme (see Section 2 and 3).

We now address the remarks of the Referee below:

1) "While it is understandable that Eq. (6) produces one dark band and two bright bands, its connection to the previous Lambda system is not clear. Here all the three bands are overlapping without the coupling As, thus it doesn't look like a Lambda-type."

As for a standard \Lambda system, where three states at different energies are coupled resonantly by time-dependent fields, we move to a rotating frame and apply the rotating wave approximation; after this operation, the three states are at the same energy (up to some detuning) and they are coupled by time-independent fields. This is precisely the situation described in Eq. (6). We now clarify this in the revised manuscript [see Eqs. 1-2 and Appendix B].

2) "What is the value of As in Fig. 2(c), and its relation to Ωs(k) (in the caption of Fig. 2)?"

We thank the Referee for this relevant question. The Ωs(k) are the matrix elements connecting the states 1,2 and 3, e.g \Omega_1^*= i <1(k) | H | 3(k)>; we now explicitly define this quantity in the revised manuscript [see Eqs. 7-8, as well as the new introductory Section 2]. Considering the multi-species configuration Hamiltonian in Eq. (6) [now Eq. 13!], which is illustrated in Fig. 2, the Ωs are set by the coupling terms (second sum in Eq. 13) and they are thus directly proportional to As in Eq. 13. The actual values of As are not very instructive (as they strongly depend on the system's detail), hence we opted to provide the maximal value of the effective couplings Ωs instead (which directly enter the common lambda-system description); see caption of Fig. 2.

3) "Finally, why is the trivial band so flat? If naively H_HH with phi=0 is taken, the bandwidth is \sim J. Is it due to the fact that only every q sites are used along the x-axis, which implies Jx \ll Jy?"

We also thank the Referee for this question. The trivial band is taken from a very deep state-dependent lattice, whose bandwidth is assumed to be negligible as compared to all other energy scales. We emphasize that this bandwidth is not related to the parameter J entering the Hofstadter model in Eq. (5), as it stems from a different (state-dependent) lattice. We now fully clarify this point in the revised version (above Eq. 13).

We believe that our revised manuscript takes all the suggestions of the Referee into account, and that it is now in a suitable form for publication.

%%%%%%%%%%%%%%%%%%%%%%%%%%%

Reply to Referee 3:

We thank Referee 3 for their careful reading of our work. We are very pleased that the Referee finds our results new and interesting.

The Referee's main criticism concerned the fact that there was "very little intuition behind the main result " (i.e. the sum rule CD=C1+C2-C3). This relevant criticism motivated us to develop an analytical proof of this sum rule, which we now proudly present in our revised manuscript. This mathematical proof, which is based on the introduction of a well-defined homotopy (see Eqs. 9 to 12), represents a major improvement and extension of our manuscript.

In particular, the addition of this proof simultaneously and rigorously addresses several questions and criticisms of the Referee. For instance, this proof indicates that the sum rule still holds even if the light-coupling carries angular momentum; it explains why the third state contributes to the Chern number, and why the phase between states 1 and 2 does not contribute.

Regarding the question "why do we need three levels":

First of all, we emphasize that our construction entirely relies on the notion of the dark state, which allows to generate a flat dark band with predictable Chern number (as dictated by the simple sum rule involving the bare bands’s Chern numbers). Our dark-state construction requires at least three bands, by definition of the dark state.

That being said, one can study the fate of two bare bands (1,2), resonantly coupled by some driving fields; this configuration would lead to two dressed bands (\pm), and the Chern numbers would satisfy (C+) + (C-) = C1 + C2, whenever the dressed bands are separated by a gap (i.e. whenever the Chern numbers are well defined). Now, a simple calculation shows that considering two bare bands with different Chern numbers (C1 \ne C2) naturally leads to a gap closing point within the Brillouin zone (for a drive that is sufficiently strong to potentially change the initial Chern numbers); hence this situation cannot be considered to generate dressed bands with tunable Chern numbers. One is thus left with the scenario where C1=C2. We note that C1=C2=0 is actually the starting point for the construction of Haldane-type models through Floquet engineering (see, for instance, the experiments in the group of Sengstock and Weitenberg in Hamburg). However, in that case, it is not possible to: (a) simply design a flat dressed band, and (b) to offer an intuitive prediction for the dressed band's Chern numbers for a general driving scheme. This motivates the use of the dark-state approach (using more than two bands), as explained above. We now further motivate our approach in the abstract and introduction.

Regarding the application to the five-state M-scheme: a similar proof can indeed be obtained for the sum-rule formula, which would indeed take the form proposed by the Referee: CD=C_1+C_3+C_5-C_2-C_4. This result is based on the observation that the four bright states can be expressed in a form similar to Eq. (4); applying a similar homotopy argument as for the 3-state system, one obtains CD=(C1+...+C5) - 2C2 - 2C4 = C1+C3+C5-C2-C4.

Finally, the Referee suggested that our work "seems very similar to Panas2020". This remark motivated us to fully clarify -- in the manuscript -- the main and crucial differences between the results of Panas2020 [Ref. 63] and our work (see the last paragraph of Section 3). We hereby reproduce this added paragraph:

"We note that similar sum rules were numerically investigated in the multilayer configuration of Ref. [63]. The latter work considered the stacking of trivial (boron-nitride-type) and non-trivial (Haldane-type) 2D lattices, and it explored the total Chern number of the system at half-filling as a function of the inter-layer coupling strength. In that context, the calculated Chern number reflects the topology of a three-fold degenerate band, which is associated with the three underlying layers; using an open-system approach, this total Chern number was decomposed as a sum of sub-system indices [64]. We point out that the approach developed in the present manuscript is radically different, as it involves the Chern number of a non-degenerate and well-isolated dark band; in particular, the sum rule in Eq. (9) relates the dark band’s Chern number to the Chern numbers of the individual bare bands, without requiring the use of an open-system formalism."

Regarding the question "what makes HH bands reminiscent of the Landau levels?": HH bands can be (nearly) flat and simultaneously have a Chern number equal to one.

We believe that our revised manuscript takes all the suggestions of the Referee into account, and that it is now in a suitable form for publication.

---

## Round 4 · List of Changes

List of changes:

- The abstract has been revised so as to better highlight the asset of our dark-state scheme, namely, the possibility of realizing nearly flat Bloch bands with *predictable* and tunable Chern number.

- Section 2 has been revised in view of providing a more pedagogical introduction to the Lambda setting; we also provide a more explicit definition of the Lambda system of Bloch bands in Section 3 [see Eqs. 7 and 8].

- The most important revision concerns the addition of the mathematical proof of the sum rule [Eq. 9], which constitutes the central result of our work; see Eqs. 9-12 and related text.

- A clear comparison with Panas2020 [Ref. 63] is provided at the end of Section 3.

- Section 5 presents new numerical simulations of the loading and detection scheme, which are more compatible with realistic experimental time scales; see Figure 3, Table 1 and related text.

---

## Editorial Decision

publication_decision_taken:_accept